# Structure and activation mechanism of the BBSome membrane protein trafficking complex

**Sandeep K Singh[1], Miao Gui[1], Fujiet Koh[1†], Matthew CJ Yip[2], Alan Brown[1]\***

[1]Department of Biological Chemistry and Molecular Pharmacology, Blavatnik Institute, Harvard Medical School, Boston, United States; [2]Department of Cell Biology, Blavatnik Institute, Harvard Medical School, Boston, United States

**Abstract** Bardet-Biedl syndrome (BBS) is a currently incurable ciliopathy caused by the failure to correctly establish or maintain cilia-dependent signaling pathways. Eight proteins associated with BBS assemble into the BBSome, a key regulator of the ciliary membrane proteome. We report the electron cryomicroscopy (cryo-EM) structures of the native bovine BBSome in inactive and active states at 3.1 and 3.5 Å resolution, respectively. In the active state, the BBSome is bound to an Arf-family GTPase (ARL6/BBS3) that recruits the BBSome to ciliary membranes. ARL6 recognizes a composite binding site formed by BBS1 and BBS7 that is occluded in the inactive state. Activation requires an unexpected swiveling of the β-propeller domain of BBS1, the subunit most frequently implicated in substrate recognition, which widens a central cavity of the BBSome. Structural mapping of disease-causing mutations suggests that pathogenesis results from folding defects and the disruption of autoinhibition and activation.

**\*For correspondence:**
alan_brown@hms.harvard.edu

**Present address:** †Thermo Fisher Scientific, Eindhoven, Netherlands

## Introduction

Most eukaryotic cells have a solitary primary cilium capable of sensing both internal and external stimuli (*Singla and Reiter, 2006*). To achieve sensitivity, cilia segregate and concentrate components of signal transduction pathways (*Pala et al., 2017*). The prototypical example of this concentrating effect is the crowding of rhodopsin within the elaborately modified ciliary membranes of retinal photoreceptor neurons (*Sung and Chuang, 2010*). Furthermore, the transport of signaling proteins in and out of the cilium allows spatial control of signal transduction, as seen in the anti-correlated movement of Patched and Smoothened during Hedgehog signaling (*Rohatgi et al., 2007*).

The establishment of signaling pathways within cilia relies on three processes: trafficking of proteins to cilia from the cytoplasm, selective passage through a diffusion barrier known as the transition zone at the base of the cilium (*Reiter et al., 2012*), and a cilium-specific internal transport mechanism known as intraflagellar transport (IFT) (*Kozminski et al., 1993*). The BBSome (an octameric complex of BBS1, BBS2, BBS4, BBS5, BBS7, BBS8, BBS9, and BBS18 [*Jin et al., 2010*; *Loktev et al., 2008*]) was initially implicated in the import of transmembrane proteins into the cilium as it directly binds cytosolic ciliary targeting sequences of transmembrane proteins (*Berbari et al., 2008a*; *Jin et al., 2010*), it is enriched at the transition zone (*Blacque et al., 2004*; *Dean et al., 2016*), and migrates bidirectionally during IFT with the IFT-A and IFT-B complexes (*Lechtreck et al., 2009*; *Liew et al., 2014*; *Ou et al., 2005*; *Williams et al., 2014*). Furthermore, ciliary G-protein coupled receptors (GPCRs) including rhodopsin (*Abd-El-Barr et al., 2007*; *Nishimura et al., 2004*), somatostatin receptor 3 (SSTR3) (*Berbari et al., 2008b*) and neuropeptide Y receptor (*Loktev and Jackson, 2013*) were mislocalized in mice lacking BBSome subunits. Trafficking of non-GPCRs, including the polycystic kidney disease ion channel polycystin-1, was also affected (*Su et al., 2014*). However, other transmembrane proteins accumulate in BBSome-deficient cilia (*Domire et al., 2011*;

*Lechtreck et al., 2013*) including those not normally destined for cilia (*Datta et al., 2015*). This led to a model in which the BBSome promotes retrieval and export of specific transmembrane proteins from the cilium (*Nachury, 2018*) and the IFT-A complex promotes entry (*Mukhopadhyay et al., 2010*) (*Hirano et al., 2017*). Studies have implicated the BBSome in the exit of phospholipase D (*Lechtreck et al., 2013*) and SSTR3 and Smoothened (*Ye et al., 2018*) from the cilium. Thus, the BBSome is a key regulator of the composition of transmembrane proteins in the ciliary membrane, and is thought to be evolutionarily related to other transmembrane protein trafficking complexes including clathrin coats and the COPI and COPII coatomers (*Jin et al., 2010*; *van Dam et al., 2013*).

Mutations in BBSome subunits are associated with Bardet-Biedl syndrome (BBS), a ciliopathy characterized by obesity, neurocognitive impairment, postaxial polydactyly, renal anomalies, and retinal dystrophy (*Green et al., 1989*). The disruption of the spatial organization of cilia-dependent signaling pathways may underpin many of these diverse phenotypes, including retinal degeneration (*Zhang et al., 2013*) and obesity (*Guo and Rahmouni, 2011*).

The recruitment of the BBSome to ciliary membranes (where it binds transmembrane protein substrates) is mediated by a highly specific interaction with ARL6 (also known as BBS3) (*Jin et al., 2010*). ARL6 is a cilium-specific (*Fan et al., 2004*) member of the Arf family of small GTPases, which have amphipathic N-terminal helices that associate with membranes in a GTP-dependent manner (*Gillingham and Munro, 2007*). ARL6 directly regulates the entry of the BBSome into cilia, as shown by an 8-fold reduction in BBSome-positive cilia following siRNA-mediated knockdown of endogenous ARL6 (*Jin et al., 2010*). The interaction between the BBSome and ARL6:GTP has been mapped to the N-terminal β-propeller domain of BBS1 (BBS1$^{\beta prop}$), and a crystal structure of this complex using recombinant proteins from *Chlamydomonas reinhardtii* has shown that ARL6 binds blades 1 and 7 of BBS1$^{\beta prop}$ (*Mourão et al., 2014*).

Structural information for the BBSome has recently become available in the form of negative-stain reconstructions of recombinant subcomplexes (*Klink et al., 2017*; *Ludlam et al., 2019*) and a mid-resolution (4.9 Å) cryo-EM reconstruction of the complete native bovine BBSome (*Chou et al., 2019*). The latter study revealed the overall architecture of the BBSome with chemical crosslinking and cutting-edge integrated modeling approaches used to place individual subunits. One of the surprising revelations of this structure was that the BBSome was in a closed conformation incompatible with the BBS1$^{\beta prop}$:ARL6:GTP crystal structure (*Mourão et al., 2014*), suggesting a conformational change, representing an activation mechanism, must occur for the BBSome to bind ARL6. However, in the absence of high-resolution structures, unanswered questions remain about the exact atomic structure of the BBSome and its relationship to vesicle coat complexes, the mechanism of activation by ARL6, and the role of disease mutations in BBS.

Here, we use single-particle cryo-EM to determine structures of the native bovine BBSome complex with and without ARL6 at 3.5 Å and 3.1 Å resolution, respectively. These structures allow unambiguous subunit assignment and atomic models to be built for each of the eight BBSome subunits. The structures reveal the mechanism of ARL6-mediated activation and provide new insights into the pathogenesis of BBS-causing mutations and the evolutionary relationship between the BBSome and other transmembrane protein trafficking complexes.

## Results

Native BBSome complexes were isolated directly from bovine retinal tissue using recombinant, FLAG-tagged ARL6:GTP as bait (*Jin et al., 2010*). Since the BBSome interacts with only the GTP-bound form of ARL6, we used a dominant negative version of ARL6 that is deficient in GTPase activity. BBSome complexes and ARL6 were eluted from the affinity column and then purified by size-exclusion chromatography. During this step, the native BBSome complexes were recovered in different fractions from ARL6:GTP, indicating dissociation of ARL6 from the BBSome. BBSome complexes lacking ARL6 were further purified by ion exchange chromatography to yield homogenous samples suitable for structural analyses (*Figure 1—figure supplement 1a*). Immediately prior to vitrifying grids for cryo-EM, the BBSome samples were mixed with a 2 × molar excess of recombinant ARL6: GTP in the pursuit of reconstituting the BBSome:ARL6:GTP complex.

Three-dimensional classification of the cryo-EM data (*Figure 1—figure supplement 1b–d*) revealed that BBSome complexes with and without ARL6 were captured. The BBSome alone was resolved to 3.1 Å resolution and the BBSome:ARL6:GTP complex to 3.5 Å resolution (*Figure 1—*

*figure supplement 1e* and *Table 1*). We also isolated BBSome complexes that lack BBS5 or BBS7 (*Figure 1—figure supplement 1d*). These rare subcomplexes (2–4% of the total dataset) may reflect native intermediates, or dissociation of the complex during purification or vitrification. Compared to the previous mid-resolution structure (*Chou et al., 2019*), our higher-resolution data allows atomic models to be built with sidechain accuracy including for previously unbuilt domains of BBS2 and BBS7. The higher-resolution data also revealed that the N-terminal β-propeller domains of BBS2 (BBS2$^{\beta prop}$) and BBS7 (BBS7$^{\beta prop}$) and the pleckstrin homology domains of BBS5 had been misplaced at lower resolution.

## Overall architecture of the BBSome

In the absence of ARL6, the eight subunits of the BBSome are arranged in two lobes that we call the head and the body (referred to as the top and base lobes by Chou and colleagues [*Chou et al., 2019*]) (*Figure 1*). The head is formed by an asymmetric heterodimer of BBS2 and BBS7, with the other six subunits forming the body. The head and body are connected by a helical neck formed from two abutting coiled coils, one from BBS2 and the other from BBS9. BBS1$^{\beta prop}$ occupies a special position in the BBSome, cradled loosely between BBS7 in the head and BBS4 in the body. The division of the BBSome into head and body lobes with BBS1$^{\beta prop}$ considered separately is based on both the physical architecture and differences in dynamics. Relative to the body, the head is more flexible and less well resolved, while BBS1$^{\beta prop}$ shows additional flexibility independent of the head

**Table 1.** Cryo-EM data collection, refinement and validation statistics.

|  | BBSome (EMD-21144) (PDB 6VBU) | BBSome:ARL6:GTP (EMD-21145) (PDB 6VBV) |
|---|---|---|
| **Data collection and processing** | | |
| Magnification | 81,000 | 81,000 |
| Voltage (kV) | 300 | 300 |
| Electron exposure (e–/Å$^2$) | 56 | 56 |
| Defocus range (μm) | −1.1 to −2.4 | −1.1 to −2.4 |
| Pixel size (Å) | 1.06 | 1.06 |
| Symmetry imposed | C1 | C1 |
| Final particle images (no.) | 152,942 | 75,201 |
| Map resolution (Å) FSC threshold | 3.1 0.143 | 3.5 0.143 |
| **Refinement** | | |
| Resolution limit set in refinement (Å) | 3.1 | 3.5 |
| Map sharpening *B* factor (Å$^2$) | −45.9 | −43.6 |
| Model composition Non-hydrogen atoms Protein residues Ligands | 30,209 3820 2 Ca$^{2+}$ | 31,676 4000 2 Ca$^{2+}$; 1 GTP |
| *B* factors (Å$^2$) Protein Ligand | 60.8 74.0 | 53.9 82.3 |
| R.m.s. deviations Bond lengths (Å) Bond angles (°) | 0.005 0.68 | 0.004 0.71 |
| Validation MolProbity score Clashscore Poor rotamers (%) | 2.02 12.0 0.2 | 2.07 12.7 0.7 |
| Ramachandran plot Favored (%) Allowed (%) Disallowed (%) | 93.5 6.5 0.0 | 92.7 7.2 0.1 |

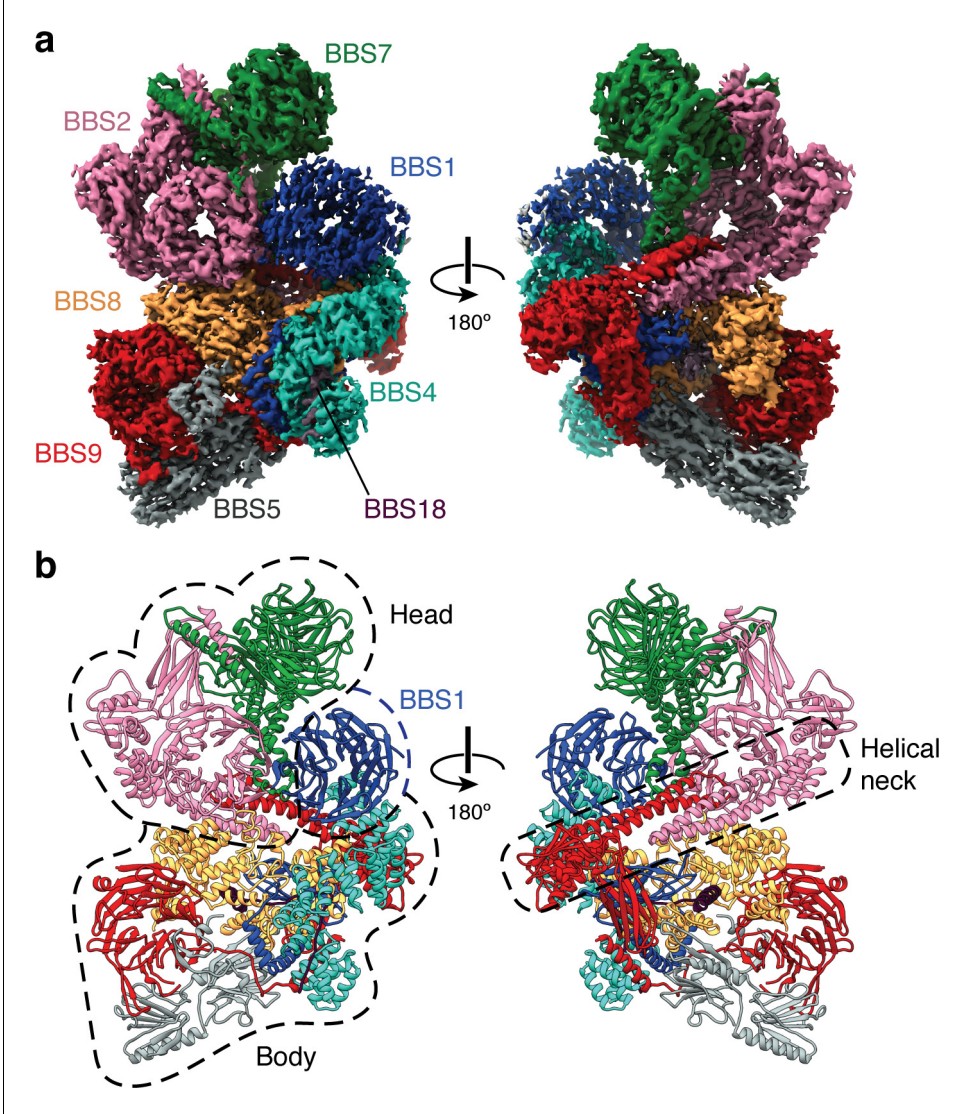

**Figure 1.** Structure of the mammalian BBSome. (**a**) Two views of the cryo-EM structure of the bovine BBSome (postprocessed map contoured at a threshold of 0.015 and colored by subunit). (**b**) Atomic models of the eight subunits of the BBSome in the same orientations as the map in panel a. The BBSome can be conceptually divided into head and body lobes (indicated with dashed lines) with the β-propeller domain of BBS1 sandwiched between. A helical neck formed from abutting coiled coils from BBS2 and BBS9 connects the head and body of the BBSome.

The online version of this article includes the following figure supplement(s) for figure 1:

**Figure supplement 1.** Cryo-EM data processing.

**Figure supplement 2.** Data quality.

movement (*Figure 1—figure supplement 1g*). To visualize the interlobe movement and generate high quality maps for model building, we used multibody refinement. The results show that the head adopts an ensemble of conformations with no single trajectory dominating (*Video 1*).

Within the body, BBS9 interacts with all other subunits of the BBSome. The extensive interconnectivity may explain why the BBSome needs three dedicated chaperonin-like BBS proteins (BBS6, BBS10, and BBS12) and CCT/TRiC family chaperonins to assemble (*Seo et al., 2010*).

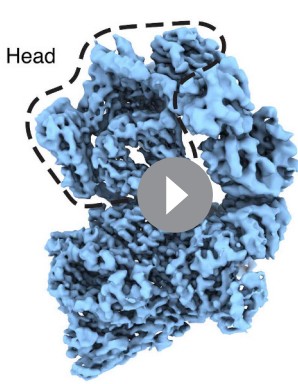

Head

**Video 1.** Dynamics of the BBSome:ARL6:GTP complex represented by the first three principal motions from multibody refinement. Two orthogonal views are shown for each motion.

https://elifesciences.org/articles/53322#video1

## BBS1, BBS2, BBS7 and BBS9 are structural homologs

BBS2, BBS7, and BBS9 all share the same five-domain architecture with an N-terminal β-propeller (βprop) followed by a heterodimerization α-helix (hx), an immunoglobulin-like GAE domain (GAE), a mixed α/β plaform (pf), and an α-helical coiled-coil (CC) (*Figure 2a*). BBS1 is a shorter homolog that lacks the last two domains. The conserved domain architectures of BBS1, BBS2, BBS7 and BBS9 suggest a common evolutionary origin. Together these four structurally homologous proteins are responsible for two-thirds of the molecular mass of the BBSome (*Table 2*), including all of the head.

The β-propeller domains of the four homologous BBS subunits are closely related members of the seven-bladed WD40 repeat family (*Figure 2b*). Each β-propeller has a 'velcro' closure with the N-terminal β1-strand serving as the outermost strand for the last blade (*Figure 2c*).

Only BBS1$^{βprop}$ contains a large structured insertion, with a helical region between residues 110 and 195 (*Figure 2d*). This insertion is one of the few regions of the BBSome for which we cannot build an accurate atomic model. Although the function of this insertion is unclear, if forms multiple chemical crosslinks with the disordered N-terminus of BBS4 (*Chou et al., 2019*), and has been suggested to bind PCM-1 at centriolar satellites during BBSome assembly (*Chou et al., 2019*; *Kim et al., 2004*).

BBS2$^{βprop}$ is unique among the BBSome β-propeller domains, as contains two Dx[D/N]xDG-like calcium-binding loops (*Rigden et al., 2011*); the first in blade 4 and the second in blade 6. A calcium cation can be seen bound to both loops, coordinated by a network of acidic sidechains and the mainchain of the conserved glycine (*Figure 2e*). A mutation (D170N) in the first of these loops is associated with BBS (*Patel et al., 2016*), suggesting that calcium binding by BBS2 is required for the proper functioning of the BBSome.

The β-propeller domains of BBS1, 2, 7 and 9 are followed by an α-helix and an immunoglobulin-like β-sandwich (*Figure 3a–c* and *Figure 3—figure supplement 1a*). This β-sandwich structurally resembles GAE domains, which are found in two different types of clathrin adaptors; the adaptin subunits of clathrin adaptor protein (AP) complexes (*Owen et al., 1999*; *Traub et al., 1999*) and the monomeric GGA family of clathrin adaptor proteins (*Dell'Angelica et al., 2000*). However, the GAE domains of the BBSome and clathrin adaptors differ in both topology − the β4 strand participates in different β-sheets (*Figure 3—figure supplement 1b–c*) − and function. Whereas the GAE domains of clathrin adaptors recruit accessory proteins to clathrin by binding hydrophobic motifs within the cytosolic tails of transmembrane proteins (*Brett et al., 2002*; *Miller et al., 2003*), the BBSome GAE domains are involved in heterodimerization. BBS2$^{GAE}$ dimerizes with BBS7$^{GAE}$ in the head (*Figure 3b*), and BBS1$^{GAE}$ dimerizes with BBS9$^{GAE}$ in the body (*Figure 3c*). The dimerization interface occludes the peptide-binding site of the clathrin adaptor GAE domains (*Brett et al., 2002*; *Jürgens et al., 2013*; *Miller et al., 2003*) (*Figure 3d*). The BBSome GAE domains also show low sequence and structural similarity with one another. For example, both BBS7$^{GAE}$ and BBS9$^{GAE}$ have a strand insertion between the β3 and β4 strands, but this additional strand contributes to different β-sheets in the two subunits (*Figure 3—figure supplement 1b*). These structural differences likely prevent incorrect pairing between BBSome subunits during assembly. The α-helix that precedes the GAE domain is part of the dimerization interface and forms a short coiled-coil with the corresponding α-helix of its partner subunit (*Figure 3a–c*).

In BBS2, BBS7 and BBS9, the GAE domain is followed by a domain which consists of a single β-sheet and two α-helices (*Figure 3—figure supplement 1d*). This domain resembles the platform domain that follows the GAE domain in the α-adaptin subunit of the clathrin AP-2 complex (*Owen et al., 1999*; *Traub et al., 1999*) but lacks the N-terminal α-helix and has an additional

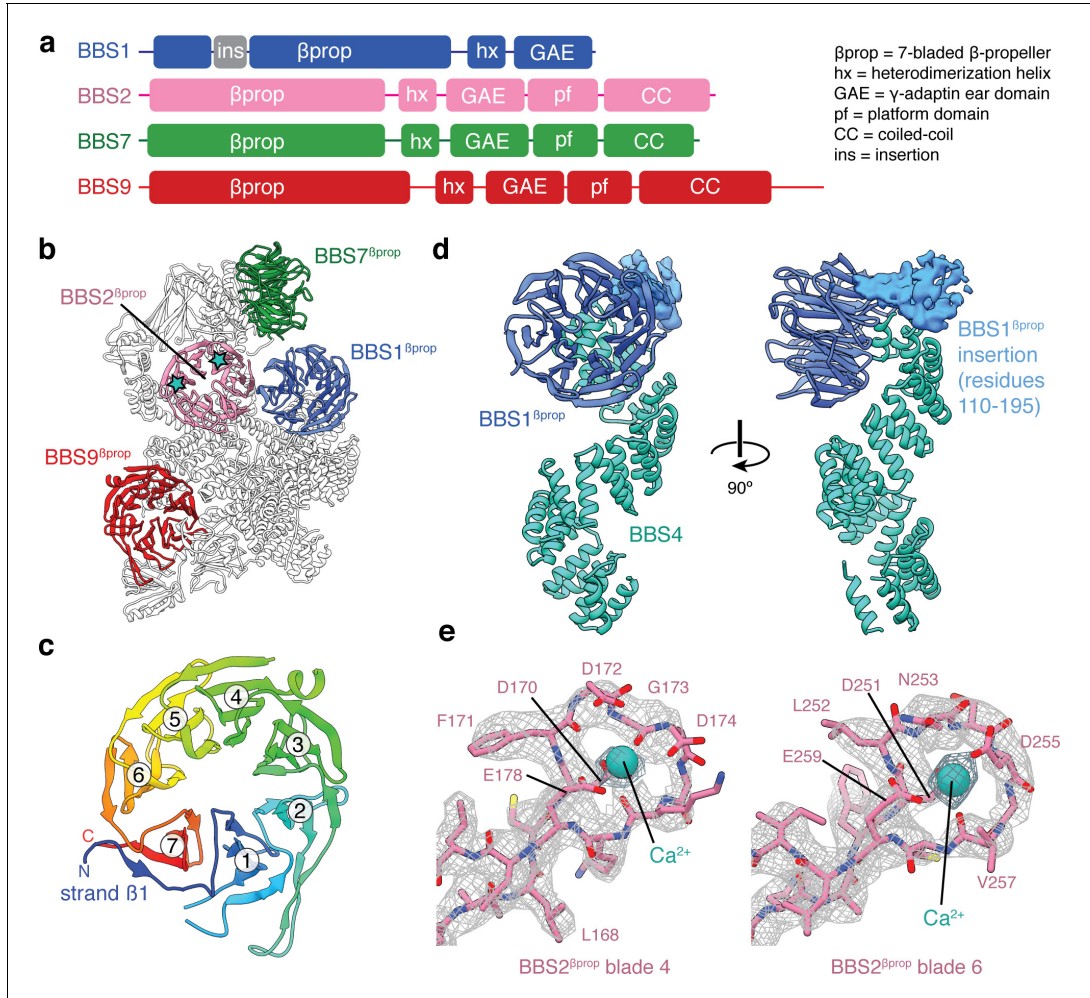

**Figure 2.** β-propeller domains of the BBSome. (**a**) Domain organization of BBS1, BBS2, BBS7, and BBS9. (**b**) The bovine BBSome contains four homologous β-propeller domains. The positions of the calcium cations that bind BBS2$^{βprop}$ are marked with a star. (**c**) BBS9$^{βprop}$ rainbow colored from N to C-terminus. The N-terminal β1-strand serves as a 'velcro' closure for blade 7. The individual blades are numbered. (**d**) BBS1$^{βprop}$ contains a helical insertion that likely interacts with the N-terminus of BBS4. (**e**) Calcium-binding loops of BBS2$^{βprop}$. Residue D170 is mutated in Bardet-Biedl syndrome (*Patel et al., 2016*).

**Table 2.** Proteins present in the BBSome or BBSome:ARL6:GTP complexes.

| Protein | NCBI accession | Protein length (residues) | Molecular mass (kDa) | Total built residues (BBSome) | Total built residues (BBSome:ARL6:GTP) |
|---------|---------------|--------------------------|---------------------|------------------------------|----------------------------------------|
| BBS1 | XP_010819476.1 | 668 | 72.9 | 486 | 486 |
| BBS2 | NP_001033249.1 | 721 | 79.8 | 659 | 659 |
| BBS4 | NP_001069424.1 | 519 | 58.2 | 386 | 391 |
| BBS5 | NP_001094602 | 341 | 38.8 | 300 | 300 |
| BBS7 | NP_001178275.2 | 715 | 80.4 | 698 | 706 |
| BBS8 | XP_024853996 | 501 | 56.6 | 475 | 475 |
| BBS9 | NP_001179782 | 887 | 99.1 | 764 | 764 |
| BBS18 | XP_003587939.1 | 69 | 8.1 | 52 | 52 |
| ARL6 | NP_001069250.1 | 186 | 21.1 | - | 167 |

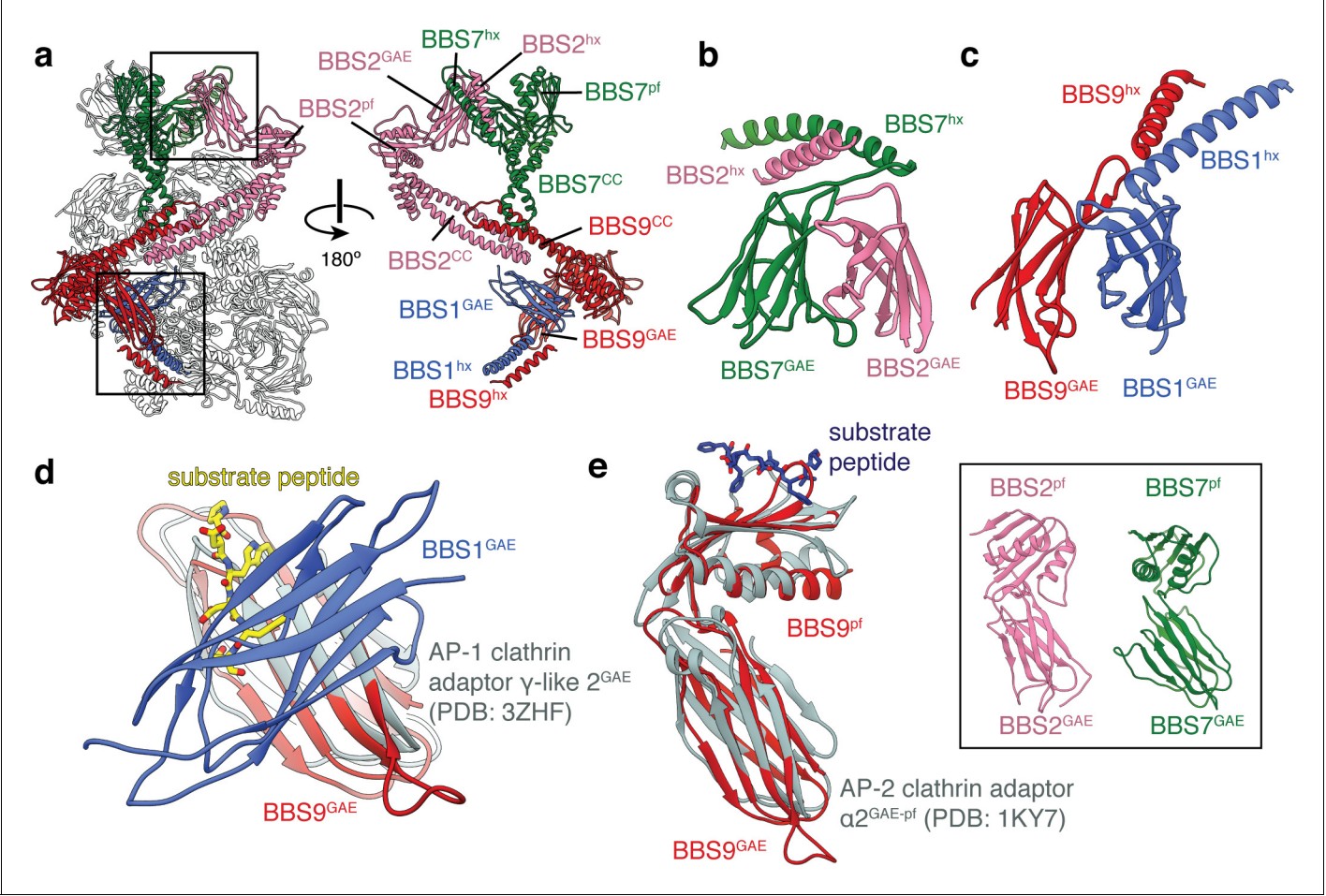

**Figure 3.** BBS1, BBS2, BBS7 and BBS9 are homologous proteins with similarities to the clathrin adaptor proteins. (a) Location of BBS1, BBS2, BBS7, and BBS9 in the BBSome, colored except for their β-propeller domains. GAE heterodimers shown in panels c and d are boxed. In the rotated view all non-colored subunits are removed for clarity. (b) Heterodimerization of BBS2 and BBS7 involves the hx-GAE module. (c) Heterodimerization of BBS1 and BBS9. (d) Superposition of BBS9$^{GAE}$ with the GAE domain of AP-1 clathrin adaptor subunit γ-like 2 reveals that the heterodimerization interface with BBS1$^{GAE}$ would occlude the substrate binding pocket. (e) The GAE-pf module of BBS2, BBS7 and BBS9 resembles the equivalent module of the AP-2 clathrin adaptor α2-adaptin. While BBS9$^{GAE-pf}$ superposes closely with α2-adaptin, the GAE and pf domains of BBS2 and BBS7 (inset) adopt different orientations relative to one another.

The online version of this article includes the following figure supplement(s) for figure 3:

**Figure supplement 1.** Topology of the BBSome GAE and platform (pf) domains.

C-terminal β-strand (*Figure 3—figure supplement 1e–f*). Like the GAE and platform domains of α-adaptin, the GAE and platform domains of the BBSome subunits make extensive, hydrophobic contacts with one another (with an interface of 520–610 Å$^2$). The relative orientation between the GAE and pf domains of BBS9 mirrors those of α-adaptin (*Owen et al., 1999*; *Traub et al., 1999*) and the modules are closely superimposable (*Figure 3e*). However, the same domains in BBS2 and BBS7 adopt different orientations relative to one another (*Figure 3e*, inset), which prevents the modules being readily superimposable with α-adaptin. Like the GAE domain, the platform domain of α-adaptin is capable of binding substrate peptides through a hydrophobic pocket (*Brett et al., 2002*). In the BBSome, these platform domains are solvent accessible, but are yet to be implicated in substrate recognition.

In all three subunits, the platform domain is followed by a helical C-terminal region containing a coiled-coil. The coiled-coils of BBS2 and BBS9 come together to form the neck of the BBSome (*Figure 1b*). The coiled-coil of BBS7 is unpaired but contacts the midpoint of the neck (*Figure 3a*).

## Domain organization of the body

BBS4, BBS5, BBS8 and BBS18 make up the remaining third of the molecular mass of the BBSome. BBS4 and BBS8 are related proteins with tetratricopeptide repeats (TPRs) that fold into α-solenoids (*Figure 4*). BBS8 occupies a central region of the body whereas BBS4 runs along the side (*Figure 4b*). The two subunits are physically connected with the C-terminus of BBS8 binding perpendicular to the midsection of BBS4 (*Figure 4c*). Whereas BBS4 forms a conventional uninterrupted α-solenoid, BBS8 has an insertion between the third and fourth α-helices of its α-solenoid (residues 48–158). This insertion consists of two short α-helices and long loops that fold together into a compact domain (*Figure 4b–c*). The density for this region is considerably weaker than for the neighboring environment, suggesting it is flexible or capable of unfolding. TPR-containing proteins typically bind a specific linear peptide within the concave surface of the α-solenoid (*Zeytuni and Zarivach, 2012*). In the case of the BBS4 and BBS8, the linear peptide is BBS18, the smallest BBSome subunit (*Figure 4c*). By stretching between BBS4 and BBS8, BBS18 appears to stabilize their association. In absence of BBS18, BBS4 fails to incorporate into the BBSome (*Loktev et al., 2008*).

BBS5, the remaining subunit, is located at the periphery of the body in extensive contact with the edge of BBS9$^{\beta prop}$ (*Figure 5a*). BBS5 contains tandem pleckstrin homology (PH) domains (BBS5$^{N-PH}$ and BBS5$^{C-PH}$) and an extended C-terminus that forms additional interactions with BBS9. Each PH domain consists of two curved antiparallel β-sheets forming a hydrophobic gorge capped by an amphiphilic α-helix. Despite just 25% sequence identity, the two domains share remarkable structural similarity with a root-mean-square deviation (r.m.s.d.) of 1.0 Å (*Figure 5b*). Many PH domains interact with the negatively charged headgroups of phosphoinositides through an electropositive binding pocket at the apex of the domain (*Ferguson et al., 2000*). As recombinant full-length BBS5 and BBS5$^{N-PH}$ have been shown to interact with phosphatidic acid and phosphoinositides in lipid-protein overlay assays (*Nachury et al., 2007*), we examined whether the BBS5 PH domains had retained this pocket (*Figure 5c*). We also considered the noncanonical phosphoinositide binding site of the PH-family GLUE domains from ESCRT complexes (*Teo et al., 2006*), which are structurally

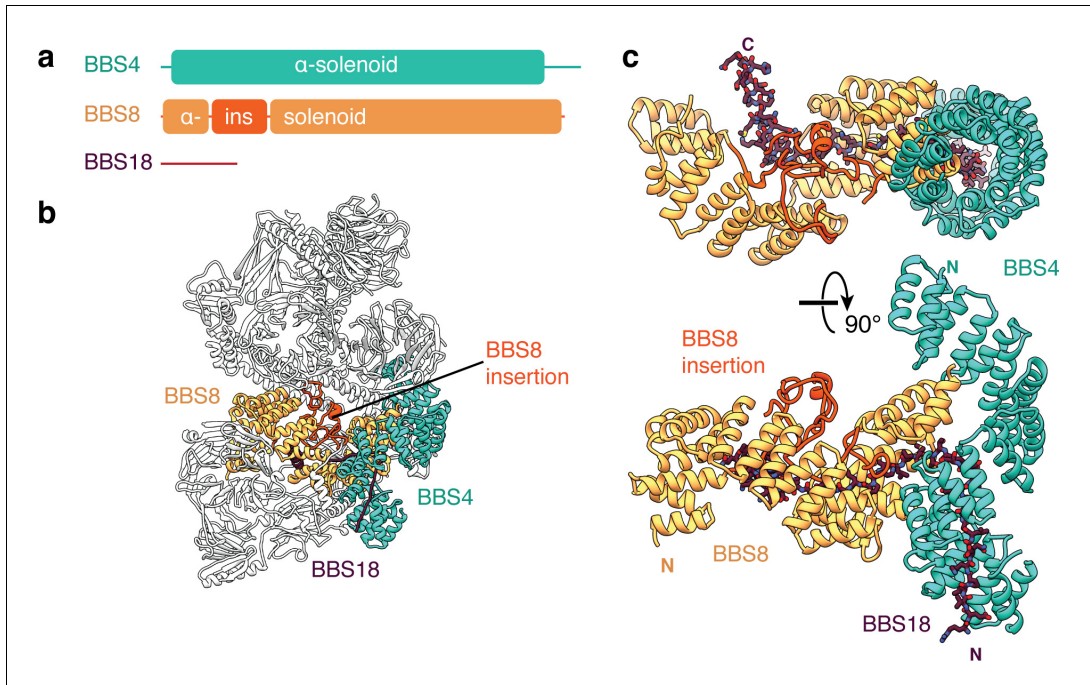

**Figure 4.** BBS18 spans the α-solenoids of BBS4 and BBS8. (a) Domain organization of BBS4, BBS8 and BBS18. BBS18 is 69 residues long and does not form a globular domain. BBS8 has an insertion (ins.) between tetratricopeptide repeats 2 and 3. (b) Location of BBS4, BBS8 and BBS18 in the BBSome. (c) Two views of the BBS4-BBS8-BBS18 subcomplex. BBS8 binds perpendicular to BBS4. The insertion in BBS8 forms a globular domain made from long loops and short α-helices. BBS18 binds the concave surfaces of the BBS8 and the N-terminal half of BBS4.

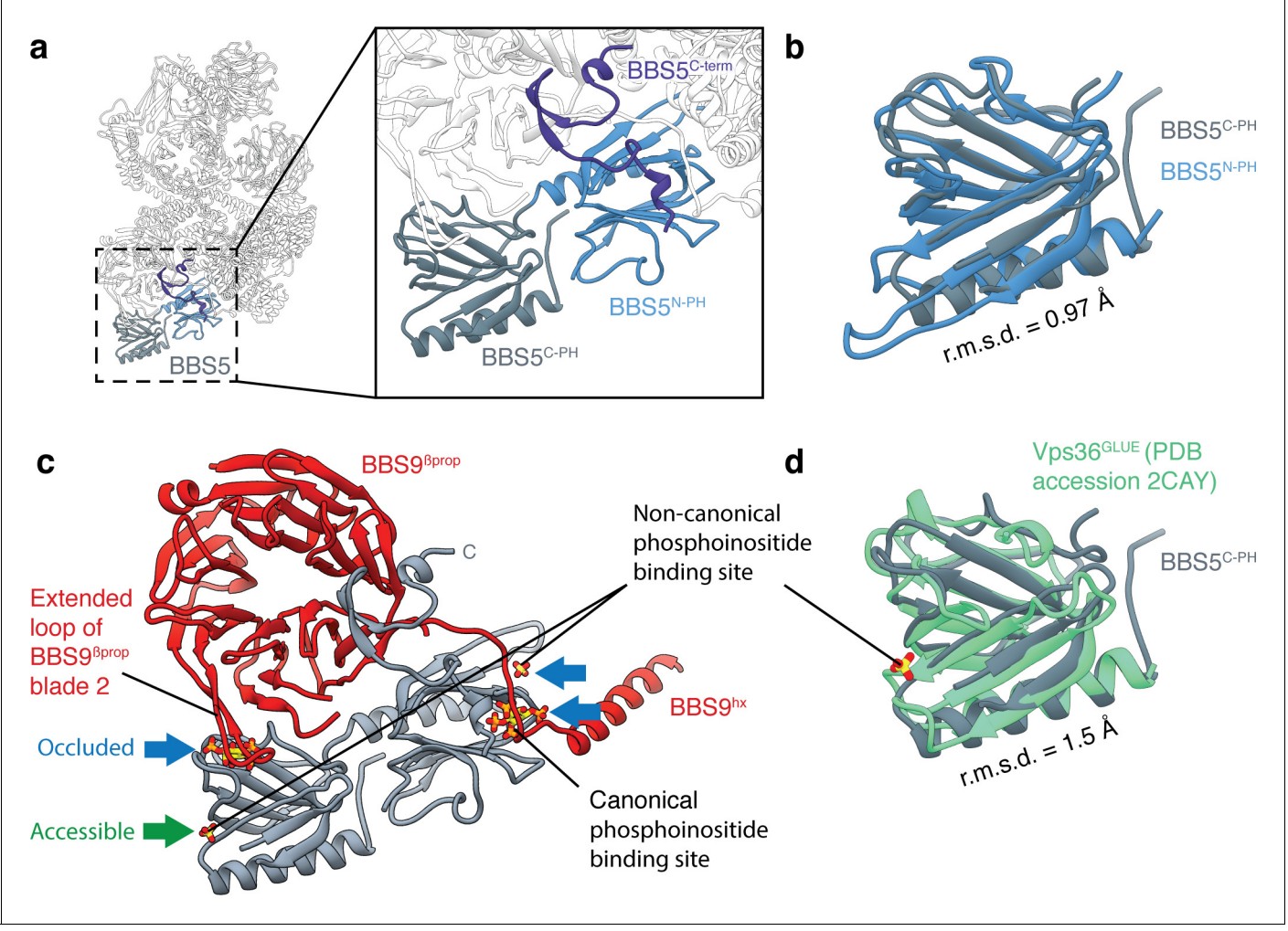

**Figure 5.** BBS5. (a) Position of BBS5 at the periphery of the BBSome body. BBS5 has tandem pleckstrin homology domains (BBS5^N-PH and BBS5^C-PH) and an extended C-terminus (BBS5^C-term). (b) BBS5^N-PH and BBS5^C-PH superpose with a root-mean-square deviation (r.m.s.d.) of 0.97 Å. (c) Potential phosphoinositide binding sites were determined from crystal structures of pleckstrin homology domains in complex with inositol-(1,3,4,5)-tetrakisphosphate (PDB: 1FAO) (*Ferguson et al., 2000*) or sulfate ions (PDB: 2CAY) (*Teo et al., 2006*). Three of the four potential binding sites in BBS5 are occluded by other BBSome subunits (blue arrows). The fourth (green arrow) is accessible but not conserved. (d) Superposition of BBS5^C-PH with the GLUE domain of Vps36, a component of the ESCRT-II complex (*Teo et al., 2006*). Vps36^GLUE has a non-canonical phosphoinositide binding site (identified based on the binding site of a sulfate ion).

similar to the BBS5 PH domains (*Figure 5d*). For both potential binding sites, the phosphoinositide binding sites are not conserved in either BBS5^N-PH or BBS5^C-PH. Furthermore, both pockets of BBS5^N-PH and the conventional pocket of BBS5^C-PH are occluded by elements from BBS9 (*Figure 5c*). The GLUE-specific phosphoinositide binding site of BBS5^C-PH is open to solvent, but lacks the basic residues required to bind phosphoinositides. We therefore conclude that if BBS5 does bind phosphoinositides in vivo, it is either through an unknown interface or after a conformational change that exposes the phosphoinositide binding sites.

## ARL6-mediated activation of the BBSome

The BBSome is recruited to ciliary membranes by membrane-associated, GTP-bound ARL6 (*Jin et al., 2010*). A crystal structure has shown that ARL6:GTP interacts with blades 1 and 7 of the BBS1^βprop (*Mourão et al., 2014*). However, this binding site is occluded in the BBSome structure, as blade 7 of BBS1 forms a continuous eight-stranded β-sheet with the corresponding blade of the adjacent BBS2^βprop (*Figure 6a*). The occlusion of the ARL6 binding site and the general flexibility of the head had led to suggestions that the head must open to allow ARL6 to bind (*Chou et al., 2019*).

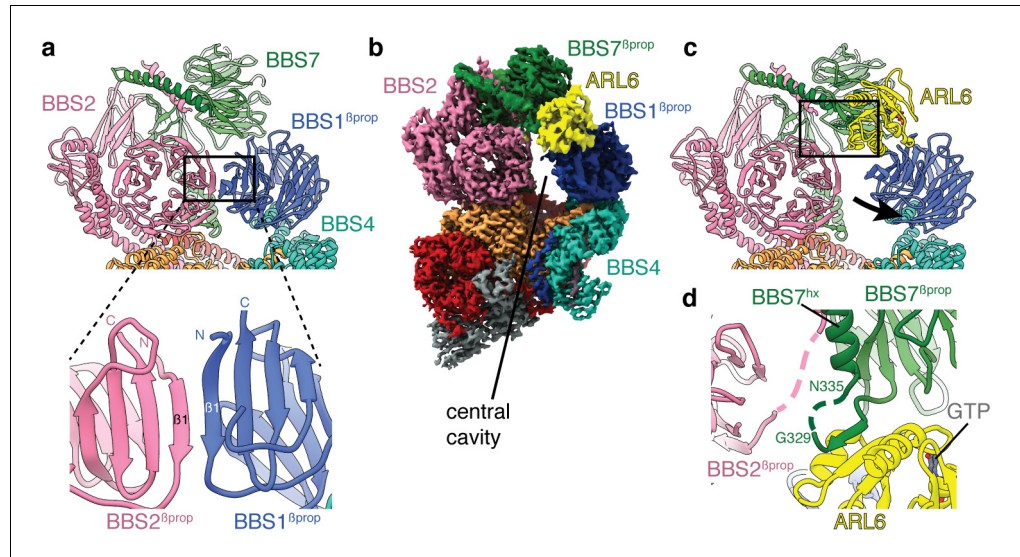

**Figure 6.** Mechanism of BBSome activation by ARL6. (**a**) In the BBSome-only state, BBS1$^{\beta prop}$ and BBS2$^{\beta prop}$ bind edge-to-edge with hydrogen bonding between their β1 strands generating a continuous eight-stranded β-sheet. (**b**) Cryo-EM structure of the BBSome:ARL6:GTP complex (postprocessed map contoured at a threshold of 0.015 and colored by subunit). ARL6 interacts with BBS7$^{\beta prop}$ and BBS1$^{\beta prop}$, which is in a rotated state compared to in the BBSome-only structure (**Figure 1**). (**c**) Rotation of BBS1$^{\beta prop}$ in the ARL6-bound state breaks the interaction with BBS2$^{\beta prop}$ and opens a central cavity in the BBSome. The region highlighted in panel d is boxed. (**d**) Details of the interaction between ARL6:GTP, BBS2, and BBS7. A loop of BBS7 that is disordered in the BBSome-only state forms a β-addition with the central β-sheet of ARL6. Regions of BBS2 and BBS7 that are not fully resolved in the cryo-EM density are shown as dashed lines.

The online version of this article includes the following figure supplement(s) for figure 6:

**Figure supplement 1.** Rotation of BBS1$^{\beta prop}$ upon ARL6 binding.

**Figure supplement 2.** Model for the assembly of the BBSome at ciliary membranes.

However, our 3.5 Å resolution structure of the BBSome:ARL6:GTP complex shows that the head remains in a closed, downward conformation even in the presence of ARL6 (**Figure 6b**). Rather, BBS1$^{\beta prop}$ swivels in its cradle between BBS4 and BBS7 to accommodate ARL6 (**Figure 6c** and **Figure 6—figure supplement 1**). This swiveling action involves a rotation of approximately 25° and a movement of 13 Å away from BBS2$^{\beta prop}$. The N-terminal half of BBS4 shows a small 2–3 Å displacement to accommodate the movement of BBS1$^{\beta prop}$. However, the hx and GAE domains of BBS1 remain static due to their attachment to BBS1$^{\beta prop}$ through a flexible linker. The swiveling of BBS1$^{\beta prop}$ opens a central cavity in the BBSome with dimensions of 50 × 15 Å, wide enough to accommodate a polypeptide chain. This cavity is flanked by the newly exposed edges of BBS1$^{\beta prop}$, BBS2$^{\beta-prop}$ and BBS7$^{\beta prop}$ as well as BBS4 and BBS8 in the body.

The interaction between ARL6:GTP and the BBSome-bound BBS1$^{\beta prop}$ is similar to that seen in the crystal structure of the *C. reinhardtii* BBS1$^{\beta prop}$:ARL6:GTP ternary complex (**Mourão et al., 2014**). The first and last blades of BBS1$^{\beta prop}$ interact with the switch two loop and helices α3 (residues 75–78) and α4 (residues 98–108) of the GTP-bound ARL6. Density for GTP (**Figure 1—figure supplement 2c**) and the ordered switch loops of ARL6 are clearly visible in our reconstruction. We also see an additional interaction between ARL6 and the loop that connects BBS7$^{\beta prop}$ with BBS7$^{hx}$ (residues 320–335) (**Figure 6d**). This linker is disordered in the BBSome-only structure but binds along the β-edge of the central β-sheet of ARL6. The corresponding linker in BBS2 also comes close to ARL6 (**Figure 6d**), although the density is insufficiently resolved to build a model of this interaction. These contacts may stabilize the BBSome:ARL6 interaction and the downward position of the head in the presence of ARL6.

## Structural mapping of BBS mutations

Taking advantage of our high-quality maps in which individual sidechains are well resolved, we mapped known disease mutations in human BBSome subunits and ARL6 onto the structure of the bovine BBSome:ARL6:GTP complex (*Figure 7a*). Pathogenic mutations were obtained from a curated list of BBS-associated mutations (*Chou et al., 2019*) supplemented with ARL6 mutations from the ClinVar database (*Landrum et al., 2014*) (*Table 3*). Only non-synonymous polymorphisms annotated as pathogenic in either BBS or retinitis pigmentosa were considered. BBS1 and BBS2 are the two most commonly mutated genes in BBS (*Forsythe and Beales, 1993*) with the majority of mutations located in their β-propeller domains. This includes the BBS1$^{M390R}$ mutation, the single most common mutation found in human BBS patients and one which is sufficient to induce BBS phenotypes including retinal degeneration and obesity in a mouse model (*Davis et al., 2007*). Our analysis suggests that many of the mutations within BBS1$^{βprop}$ and BBS2$^{βprop}$ would result in the introduction of bulky or charged residues that would disrupt hydrophobic packing and correct folding, as shown experimentally for the M390R mutation introduced into recombinant BBS1$^{βprop}$ (*Mourão et al., 2014*). The vulnerability of BBS1$^{βprop}$ and BBS2$^{βprop}$ reflects their important contributions to the BBSome's autoinhibitory and activation mechanisms. Destabilizing mutations within these domains would affect formation of the head, the positioning of BBS1$^{βprop}$, and recruitment by ARL6. We also note a cluster of mutations in BBS7 (L317V, H323R, G329V, R346Q) close to its interaction site with ARL6. In particular, H323R and G329V are within the flexible linker that only becomes ordered in the presence of ARL6 (*Figure 7b*). Mutations within this linker may disrupt ARL6-mediated BBSome recruitment.

Most mutations outside BBS1$^{βprop}$ and BBS2$^{βprop}$ can be rationalized as causing misfolding of individual subunits, predominantly by affecting the packing of the hydrophobic cores. Other mutations map to the interfaces between subunits. For example, BBS4$^{N309K}$ (*Muller et al., 2010*) maps to the interface with BBS18 (*Figure 7c*), BBS1$^{E224K}$ (*Redin et al., 2012*) maps to the interface between BBS1$^{βprop}$ and BBS4 (*Figure 7d*), and BBS2$^{R632P}$ (*Katsanis et al., 2001*) maps to the interface

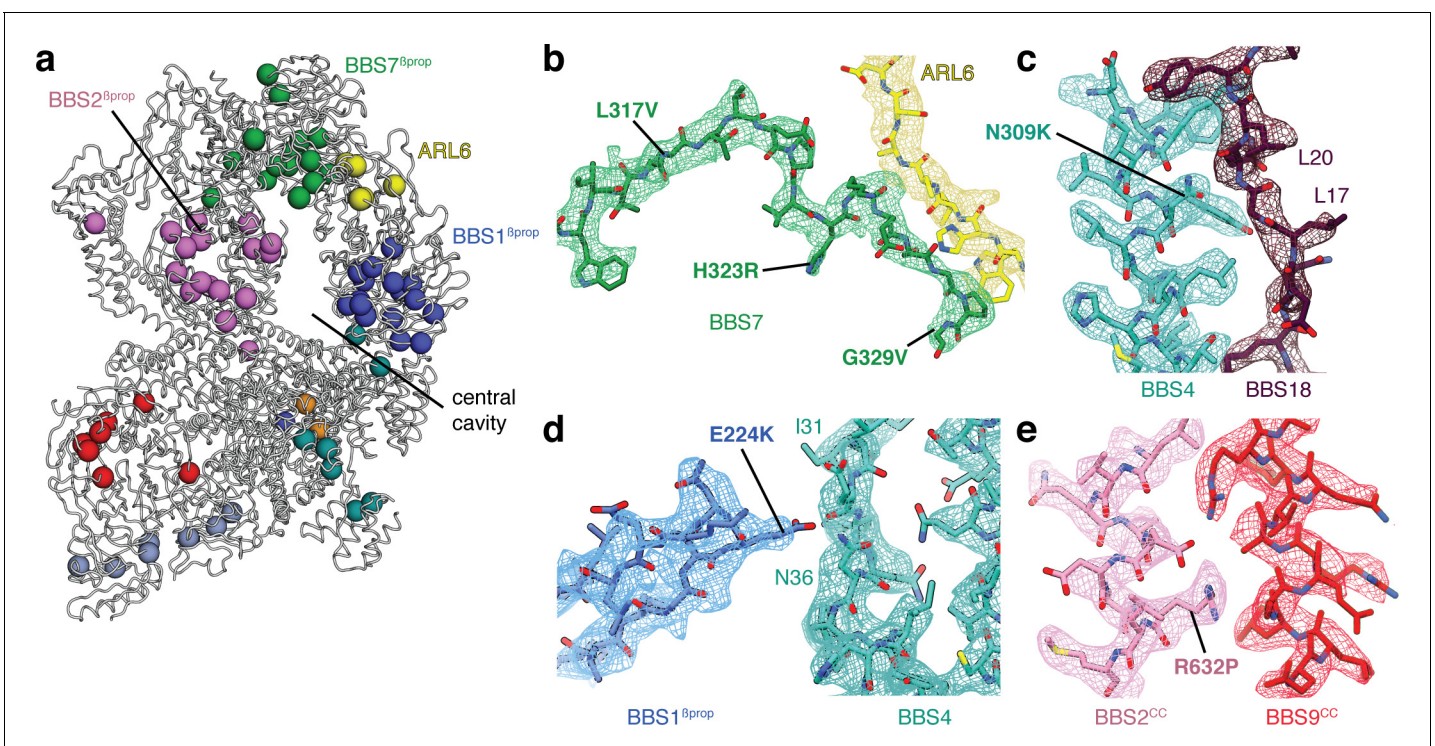

**Figure 7.** Structural insights into Bardet-Biedl syndrome. (a) Known disease-causing mutations mapped onto the model of the BBSome:ARL6:GTP complex. Each sphere, colored by subunit, represents a missense mutation associated with either BBS or retinitis pigmentosa. (b) Mutations in BBS7 close to the binding site with ARL6. (c–e), BBS mutations that could disrupt subunit association and therefore proper BBSome assembly. Panels b-e show the postprocessed map contoured at a threshold between 0.015–0.02 and colored by subunit.

**Table 3.** Mutations in ARL6 associated with BBS or retinitis pigmentosa (RP) that are mapped onto the structure in *Figure 7a*.
The mapped disease-associated mutations in core BBSome subunits are provided as a supplemental table in *Chou et al. (2019)*.

| Gene | Protein mutation | Phenotype | Reference |
|------|------------------|-----------|-----------|
| ARL6 | T31M | BBS | (*Fan et al., 2004*) |
| ARL6 | T31R | BBS | (*Fan et al., 2004*) |
| ARL6 | A89V | RP | (*Aldahmesh et al., 2009*) |
| ARL6 | I91T | BBS | (*Chandrasekar et al., 2018*) |
| ARL6 | I94T | RP | (*Khan et al., 2013*) |
| ARL6 | G169A | BBS | (*Young et al., 1998*) |
| ARL6 | L170W | BBS | (*Fan et al., 2004*) |

between the coiled-coils of BBS2 and BBS9 in the neck (*Figure 7e*). These mutations may affect the proper assembly of the BBSome.

# Discussion

## Mechanism of BBSome activation and implications for substrate recognition and IFT

Our structures of the BBSome with and without ARL6 show that activation of the BBSome at ciliary membranes requires a swiveling of BBS1$^{\beta prop}$ that widens a cavity in the body of the BBSome. This relief of autoinhibition through a conformational change is similar to other Arf-regulated systems, including the clathrin adaptor complexes (*Ren et al., 2013*). The rotation of BBS1$^{\beta prop}$ and the opening of the cavity may allow substrate recognition by newly accessible elements. In particular, the breaking of the continuous β-sheet between BBS1 and BBS2 exposes β-edge strands, which are common mediators of protein–protein interactions (*Remaut and Waksman, 2006*) that have the potential to hydrogen bond to cytosolic regions of transmembrane proteins. BBS1 is especially implicated in substrate recognition and interacts with all known substrates of the BBSome including the C-terminal cytosolic tails of Smoothened and Patched-1 (*Zhang et al., 2012*), the Leptin receptor (*Seo et al., 2009*), and polycystin-1 (*Su et al., 2014*). The plasticity of BBS1 in its loosely held cradle may allow it to subtly reorient to make optimal contacts with multiple cargoes. Some substrates bind other BBSome subunits as well as BBS1. Smoothened binds BBS4, BBS5, and BBS7 in co-transfection immunoprecipitation experiments (*Zhang et al., 2012*). Polycystin-1 interacts with BBS4, BBS5, and BBS8 in yeast two-hybrid screens (*Su et al., 2014*). Our structures show that these subunits are present on a relatively flat face of the BBSome that, based on the orientation induced by ARL6, would lie parallel to the ciliary membrane in vivo, forming a large interface for cargo binding (*Figure 6—figure supplement 2*). However, the relevance of these interactions is unclear as trafficking of polycystin-1 to cilia is only severely diminished in BBS1 knockdown cells (*Su et al., 2014*).

Assuming the flat surface of the BBSome abuts the membrane, the opposite face would be free to interact with the IFT complexes (IFT-A and IFT-B), with which the BBSome comigrates (*Lechtreck et al., 2009*; *Liew et al., 2014*; *Ou et al., 2005*; *Williams et al., 2014*). Recent data from visible immunoprecipitation experiments has mapped the interaction to BBS1, BBS2 and BBS9 of the BBSome, and IFT38 of the IFT-B complex (*Nozaki et al., 2019*). This is consistent with the copurification of BBSome subunits with endogenously tagged IFT38 in a human cell line (*Beyer et al., 2018*). Analysis of our structure shows that BBS1, BBS2 and BBS9 come together at the base of the neck where the coiled-coil domains of BBS2 and BBS9 meet the GAE dimerization domains of BBS1 and BBS9. Whether this is the sole binding site for the IFT complexes awaits further investigation, especially as other IFT-B subunits including IFT25 (*Eguether et al., 2014*), IFT27 (*Aldahmesh et al., 2014*; *Eguether et al., 2014*; *Liew et al., 2014*), IFT74 (*Lindstrand et al., 2016*) and IFT172 (*Schaefer et al., 2016*) are either genetically associated with BBS or have been associated with BBSome exit from the cilium.

## Relationship to vesicle coats

Our structures strengthen the proposed evolutionary relationship between the BBSome, clathrin coats, and the COPI and COPII coatomers (*Jin et al., 2010*; *van Dam et al., 2013*), which are all involved in transmembrane-protein trafficking. In particular, we show that the GAE-pf module of BBS2, BBS7, and BBS9 is structurally related to the same module found in the α-adaptin subunit of the clathrin adaptor complex, AP-2 (*Owen et al., 1999*; *Traub et al., 1999*) (*Figure 3e*). The mechanism of membrane-recruitment and activation of the BBSome by ARL6 is also reminiscent of the activation of clathrin AP complexes by Arf1 and Arf6 GTPases (*Paleotti et al., 2005*; *Ren et al., 2013*), in which a GTPase-induced conformational change precedes substrate recognition.

The α-solenoids, β-propellers, and PH-like domains of the BBSome also have equivalents in other membrane trafficking complexes. For example, α-solenoids and β-propellers are hallmarks of the protocoatomer family, although in clathrin and COP coatomers the α-solenoids and β-propellers are domains of the same protein (*Rout and Field, 2017*). BBS5-like PH domains are found in the ESCRT complexes (*Teo et al., 2006*), which are required for the formation and sorting of endosomal cargo proteins into multivesicular bodies.

The structural similarity with the clathrin adaptor complexes provides compelling support for the model that the BBSome is an adaptor complex, linking transmembrane proteins to the IFT-B complexes for active transport (*Liu and Lechtreck, 2018*). However, while other trafficking complexes oligomerize to form membrane deformations and vesicles (*Rout and Field, 2017*), the evidence that the BBSome can do likewise is limited. BBSome complexes incubated with full-length ARL6 can form electron-dense coats surrounding sections of liposomes but without inducing membrane deformation (*Jin et al., 2010*). In the absence of membranes, we observed no evidence of BBSome oligomerization by either size-exclusion chromatography or by electron microscopy. Further work will be needed to examine the evolutionary and functional relationship with vesicle coats and with the IFT complexes, which are also predicted to have evolved from a common progenitor (*van Dam et al., 2013*).

In summary, our structures of the BBSome with and without ARL6 reveal the intricate subunit arrangement of the BBSome and its mechanism of membrane-recruitment and activation by an Arf-family GTPase. We show that the ARL6 binding site includes contributions from BBS2 and BBS7 as well as BBS1. The swiveling of BBS1$^{\beta\text{prop}}$ to accommodate ARL6 and the resultant widening of a cavity within the BBSome will inform work to elucidate the molecular basis of substrate recognition and the relationship between the BBSome and the IFT machinery. Furthermore, our structures can help guide the design of future therapies, in particular CRISPR-based in vivo genetic engineering, aimed at curing or alleviating the effects of BBS.

# Materials and methods

### Key resources table

| Reagent type (species) or resource | Designation | Source or reference | Identifiers | Additional information |
|---|---|---|---|---|
| Gene (*Bos taurus*) | ARL6. NCBI Gene ID: 519014 | IDT | - | Codon optimized |
| Biological Sample | Bovine dark-adapted retinas | W L Lawson company (NE, USA) | - | |
| Strain, strain background *Escherichia coli* cells | BL21(DE3) | Novagen | 69450–4 | Chemically Competent cells |
| Affinity resin | Anti-Flag M2 | Sigma | A2220 | |
| Chemical compound | GTP | Sigma | G8877 | |
| Cryo grids | QUANTIFOIL R 1.2/1.3 | Electron Microscopy Sciences | Q4100AR1.3 | |
| Commercial assay or kit | Gibsons Assembly | Invitrogen | A14606 | |

*Continued on next page*

*Continued*

| Reagent type (species) or resource | Designation | Source or reference | Identifiers | Additional information |
|---|---|---|---|---|
| Sequence based reagents | Arl6_dN16 Fwd | This paper | PCR primers | GAAGTTCATGTGCTGTGTTTGG |
| Sequence based reagents | Arl6_dN16 Rev | This paper | PCR primers | ACTCCCACCCCCTTTATCATC |
| Sequence based reagents | Arl6_addHis_Fwd | This paper | PCR primers | TG GAA GTT CTG TTC CAG GGG CCC GATTACAAGGACGATGATGATAAAG |
| Sequence based reagents | Arl6_addHis_Rev | This paper | PCR primers | GAATTCTCGAGCGGCCGCCCTTA TGTCTTCACCGACTGAATC |
| Software, algorithm | serialEM | doi: 10.1038/s41592-019-0396-9 | RRID:SCR_017293 | https://bio3d.colorado. edu/SerialEM |
| Software, algorithm | MotionCor2 v.1.2.1 | doi: 10.1038/nmeth.4193 | RRID:SCR_016499 | |
| Software, algorithm | CTFFIND v.4.1.13 | doi: 10.1016/j.jsb.2015.08.008 | RRID:SCR_016732 | https://cistem.org/ctffind4 |
| Software, algorithm | RELION v.3.0.4 | doi: 10.7554/eLife.42166 | RRID:SCR_016274 | https://www3.mrc-lmb.cam.ac.uk/relion/index. php/Download_%26_install |
| Software, algorithm | Coot v. 0.9-pre | doi: 10.1107/S0907444904019158 | RRID:SCR_014222 | https://www2.mrc-lmb.cam.ac.uk/personal/ pemsley/coot/ |
| Software, algorithm | Phenix.real_ space_refine | doi: 10.1107/S2059798318006551 | RRID:SCR_014224 | https://www.phenix-online.org/ |
| Software, algorithm | UCSF Chimera v1.13.1 | doi: 10.1002/jcc.20084 | RRID:SCR_004097 | http://plato.cgl.ucsf.edu/chimera/ |
| Software, algorithm | UCSF ChimeraX v.0.9 | doi: 10.1002/pro.3235 | RRID:SCR_015872 | https://www.cgl.ucsf.edu/chimerax/ |
| Software, algorithm | PyMOL v2.3.2 | PyMOL Molecular Graphics System, Schrödinger, LLC | RRID:SCR_000305 | http://www.pymol.org/ |
| Software, algorithm | crYOLO | doi: 10.1038/s42003-019-0437-z | - | http://sphire.mpg.de/ wiki/doku.php?id= pipeline:window:cryolo |
| Software, algorithm | ResMap | doi: 10.1038/nmeth.2727 | - | http://resmap.sourceforge.net/ |
| Software, algorithm | I-TASSER | doi: 10.1186/1471-2105-9-40 | RRID:SCR_014627 | https://zhanglab.ccmb. med.umich.edu/I-TASSER |
| Software, algorithm | MolProbity v.4.3.1 | doi: 10.1107/S0907444909042073 | RRID:SCR_014226 | http://molprobity.biochem.duke.edu |
| Software, algorithm | SBGrid | doi: 10.7554/eLife.01456 | RRID:SCR_003511 | https://sbgrid.org/ |

## ARL6 cloning and purification

To isolate the BBSome complex from bovine retina, we first generated a recombinant bait protein, bovine ARL6. A synthetic, codon-optimized nucleotide sequence (Integrated DNA Technologies) encoding *Bos taurus* ARL6 with an N-terminal FLAG tag replacing the first 16 residues of ARL6 was inserted into a pSY5 vector using Gibson assembly. The pSY5 vector introduces an additional octa-histidine tag and PreScission cleavage site prior to the Flag tag. A dominant negative Q73L mutation was introduced to slow GTP hydrolysis (*Jin et al., 2010*). His8-3C-Flag-Δ16NARL6(Q73L) was expressed in *Escherichia coli* BL21(DE3) cells (Novagen) at 20°C overnight after induction with 1 mM isopropyl β-D-1-thiogalactopyranoside (Sigma) once the cells reached an optical density of 0.4–0.6 at 600 nm. The bacterial cells were collected by centrifugation at 7000 x g for 7 min. All subsequent steps were performed on ice or at 4°C. The bacterial cells were resuspended in lysis buffer (40 mM Tris pH 8.0, 150 mM NaCl, 10 mM imidazole, 5 mM MgCl$_2$, 4 mM β-mercaptoethanol, 0.05% NP-40, HALT protease inhibitor cocktail (Thermo Fischer Scientific)) and sonicated for a total of 8 min using 20 s on/20 s off cycles and 20% amplitude. The bacterial lysate was clarified using centrifugation at

40,000 x g for 40 min and loaded onto a 5 ml His-Trap column (GE Healthcare) pre-equilibrated with lysis buffer. The column was then washed with 100 ml of lysis buffer without protease inhibitors. The octahistidine tag was removed overnight by on-column digestion with human rhinovirus 3C protease which specifically recognizes the PreScission cleavage site. The cleaved Flag-Δ16NARL6(Q73L) protein (hereon in called 'ARL6') was eluted from the column with 25 ml of lysis buffer and concentrated to a final volume of 1 ml using a concentrator with a 10 kDa molecular weight cutoff (Thermo Fischer Scientific). ARL6 was purified to homogeneity using a Superdex 200 (16/60) size-exclusion chromatography column (GE Healthcare) and elutes as a single, symmetric peak. The peak fractions were pooled, concentrated to ~10 mg/ml, vitrified in 50 μl aliquots in liquid nitrogen, and stored at −80°C until further use.

### Preparation of retinal extracts

Bovine retinas were purchased from W L Lawson company (NE, USA). 50 g of bovine retinas were resuspended in lysis buffer (40 mM Tris pH 8.0, 150 mM NaCl, 250 mM sucrose, 5 mM MgCl$_2$, 4 mM β-mercaptoethanol, Halt protease inhibitor cocktail (Thermo Fischer Scientific)) and homogenized using a Tissue Tearor (BioSpec Products) for 1 min. The retinal tissue was further homogenized using 6–10 strokes of a glass Dounce homogenizer. The lysate was clarified by centrifugation at 40,000 x g for 50 min and the supernatant collected.

### Purification of the BBSome

Prior to generating the ARL6 affinity column, we incubated ~2 mg (100 μM) ARL6 with 2 mM GTP (final concentration) for 1 hr. The ARL6:GTP complex was then loaded onto 3 ml of anti-Flag M2 affinity resin (Sigma). The resin was washed with 30 ml of buffer + 100 μM GTP to remove any excess, unbound ARL6. Immediately before loading onto the column, 100 μM GTP (final concentration) was added to the clarified lysate. The retinal tissue lysate was loaded onto anti-Flag M2 pre-saturated with bovine of ARL6 and incubated for 1 hr at 4°C. The lysate was passed over the column using a peristaltic pump multiple times with a flow rate of 2 ml/min. Resin was washed with 40 ml lysis buffer + 100 μM GTP. The BBSome:ARL6 complex was eluted from the column with a total of 10 ml of 0.1 mg/ml Flag peptide (Sigma). Elution was performed in five steps, in which each step involved a 30 min incubation with Flag peptide. The eluted BBSome:ARL6 complex was concentrated to 500 μl using a concentrator with a 100 kDa molecular weight cutoff (Thermo Fischer Scientific) and injected onto a Superdex 200 (16/600) size-exclusion chromatography column (GE Healthcare) equilibrated with 20 mM Hepes pH 7.5, 220 mM NaCl, 5 mM MgCl$_2$, 4 mM β-mercaptoethanol. The peak fractions were pooled and concentrated using 100 kDa cut-off concentrator (Thermo Fischer Scientific) to ~0.5–0.7 mg/ml. ARL6 dissociates from the BBSome during size-exclusion chromatography. The BBSome-containing fractions were then buffer exchanged into 20 mM Hepes pH 7.5, 20 mM NaCl, 5 mM MgCl$_2$, 4 mM β-mercaptoethanol and loaded onto a 1 ml MonoQ anion exchange chromatography column (GE Healthcare). After washing with 10 column volumes of buffer, a gradient of 20 mM to 1 M NaCl was applied to elute the BBSome. The purity of the BBSome is shown in *Figure 1—figure supplement 1a*.

### Sample preparation for cryo-EM

Prior to making grids, 0.7 mg/ml BBSome (~18 μM) was mixed with 2 × molar excess of ARL6 (36 μM) and 1 mM GTP and incubated for an hour at 4°C. During incubation, holey carbon R1.2/1.3 grids with gold 400 mesh (Quantifoil Micro Tools) were glow discharged at 15 mA for 30 s (PELCO easiGlow Glow Discharge Cleaning System). 3 μl of BBSome:ARL6:GTP complexes were applied to each glow-discharged grid. Grids were blotted for 2 s with a −2 offset at ~100% humidity and 20°C before being plunge-frozen in liquid ethane using a Vitrobot Mk II (Thermo Fisher Scientific).

### Cryo-EM data collection

The grids were imaged on a Titan Krios microscope (Thermo Fisher Scientific) operating at an acceleration voltage of 300 kV and equipped with a BioQuantum K3 Imaging Filter (slit width 25 eV). Images were recorded on a K3 Summit direct electron detector (Gatan) operated in counting mode (*Figure 1—figure supplement 1b*). For data collection, we used a spot size of 4, a C2 aperture of 50 μm, and a nominal magnification of 81,000 x, yielding a pixel size of 1.06 Å. The total exposure

time of each movie stack was 4 s fractionated into 50 frames with a total exposure of approximately 56 electrons/Å$^2$. The defocus targets were −1.1 to −2.4 μm. In total, 9408 micrographs were collected from two sessions. SerialEM was used for data collection (*Schorb et al., 2019*).

## Image processing

We used MotionCor2 to correct for global and local (5 × 5 patches) beam-induced motion and to dose weight the individual frames (*Zheng et al., 2017*). CTFFIND-4.1 was used to estimate parameters of the contrast transfer function (CTF) (*Rohou and Grigorieff, 2015*). Particles were picked from the micrographs using crYOLO (*Wagner et al., 2019*) and their coordinates exported to RELION-3.0 (*Zivanov et al., 2018*) for all subsequent processing steps. Particles were extracted with a box size of 320 pixel. A single round of two-dimensional classification was performed and well-defined classes corresponding to BBSome particles were selected (*Figure 1—figure supplement 1c*). An initial map for the BBSome was generated using RELION's implementation of the stochastic gradient descent algorithm using default parameters and a mask diameter of 280 Å. The initial map was used as a reference for three-dimensional refinement. After refinement, CTF refinement and Bayesian polishing were performed. The particles from the two data collection sessions were combined after Bayesian polishing and 3D classification (without alignment) was performed (*Figure 1—figure supplement 1d*). The two best classes (based on occupancy and map quality) were selected and refined together. As this map is generated from BBSome particles with and without ARL6, we next performed focused classification with signal subtraction (FCwSS) with a mask centered on ARL6 to separate the different species. Classes with and without ARL6 were independently selected and refined. After post-processing in RELION-3.0, including correcting for the modulation transfer function of the K3 Summit direct electron detector, the resolution of the BBSome reconstruction was 3.1 Å and the resolution of the BBSome:ARL6 complex was 3.5 Å based on the FSC = 0.143 criterion (*Rosenthal and Henderson, 2003*) (*Figure 1—figure supplement 1e*). Final reconstructions were sharpened using automatically estimated B-factors (*Rosenthal and Henderson, 2003*). Local resolution calculations were performed with ResMap (*Kucukelbir et al., 2014*).

To further improve the map density of the BBSome, we used multibody refinement with masks covering the body (mask 1) and head (mask 2) lobes. BBS1$^{\beta prop}$ was included in the body mask. The masks were made in RELION with a raised-cosine soft edge. The quality of the map for the body was minimally improved with the resolution remaining unchanged at 3.1 Å, but the quality of the map for the head improved, with a nominal resolution of 3.4 Å. A third mask centered on the ARL6: BBS1$^{\beta prop}$ subcomplex was used for multibody refinement of the BBSome:ARL6:GTP complex. These masks resulted in final resolutions of 3.3 Å for the body, 3.8 Å for the head, and 4.0 Å for the ARL6: BBS1$^{\beta prop}$ subcomplex. The masked maps from multibody refinement were resampled to the pre-multibody reference and merged by taking the maximum density value at each voxel using the *vop maximum* command in Chimera (*Pettersen et al., 2004*). These chimeric maps were used for model building to take advantage of the improved map quality. Chimera was also used to generate a movie (*Video 1*) showing the motion of the lobes and the ARL6:BBS1$^{\beta prop}$ subcomplex represented by the first three eigenvectors.

## Model building and refinement

Amino acid sequences for the *Bos taurus* BBSome subunits were obtained from the NCBI (*Table 2*) and used as the input to generate comparative models with I-TASSER (*Zhang, 2008*). These models were trimmed to remove unstructured or poorly predicted regions. For BBS9$^{\beta prop}$, BBS1$^{\beta prop}$ and ARL6, the crystal structures of human BBS9$^{\beta prop}$ (PDB: 4YD8) (*Knockenhauer and Schwartz, 2015*) and *Chlamydomonas reinhardtii* BBS1$^{\beta prop}$:ARL6:GTP complex (PDB: 4V0N) (*Mourão et al., 2014*) were used directly and mutated to the *Bos taurus* sequence. The models were then placed into the BBSome density map using the fit-to-map procedure in Chimera (*Pettersen et al., 2004*) or manually in Coot v0.8.9 (*Brown et al., 2015*). These homology and crystal structures were used as starting points for model building, but most required comprehensive remodeling. All GAE, pf, and CC domains were built de novo. The previous model of the BBSome obtained by integrative modeling (PDB-Dev accession PDBDEV_00000018) (*Chou et al., 2019*) was not available or used during the modeling process. During model building and real-space refinement in Coot, torsion, planar peptide and Ramachandran restraints were used. The models were refined using Phenix.real_space_refine

(*Afonine et al., 2018*) against the composite maps from multibody refinement. During refinement the resolution limit was set to match the resolution determined using the FSC = 0.143 criterion. Secondary structure, Ramachandran and rotamer restraints were applied during refinement. Round of manual model correction in Coot was performed between rounds of refinement. The final models were validated using MolProbity v.4.3.1 (*Chen et al., 2010*) with model statistics provided in *Table 1*. FSC curves calculated between the models and the unsharpened maps are shown in *Figure 1—figure supplement 1f*.

### Figures
Figure panels were generated using PyMOL (*DeLano, 2002*), Chimera (*Pettersen et al., 2004*), or ChimeraX (*Goddard et al., 2018*). Maps colored by local resolution (*Figure 1—figure supplement 1g*) were generated with unsharpened density maps using ResMap (*Kucukelbir et al., 2014*).
Software used in the project were installed and configured by SBGrid (*Morin et al., 2013*).

## Acknowledgements

Cryo-EM data were collected at the Harvard Cryo-Electron Microscopy Center for Structural Biology. We thank S Sterling, R Walsh, and Z Li for microscopy support, S Rawson and SBGrid for computing support, and T Walton for comments.

## Additional information

### Competing interests
Fujiet Koh: is affiliated with Thermo Fisher Scientific. The author has no financial interests to declare. The other authors declare that no competing interests exist.

### Funding

| Funder | Author |
| --- | --- |
| Pew Charitable Trusts | Alan Brown |
| International Retinal Research Foundation | Alan Brown |
| E. Matilda Ziegler Foundation for the Blind | Alan Brown |
| Richard and Susan Smith Family Foundation | Alan Brown |

The funders had no role in study design, data collection and interpretation, or the decision to submit the work for publication.

### Author contributions
Sandeep K Singh, Data curation, Formal analysis, Investigation, Methodology, Writing - review and editing; Miao Gui, Fujiet Koh, Matthew CJ Yip, Investigation, Writing - review and editing; Alan Brown, Conceptualization, Formal analysis, Supervision, Funding acquisition, Validation, Investigation, Visualization, Methodology, Writing - original draft, Writing - review and editing

### Author ORCIDs
Matthew CJ Yip (ID) http://orcid.org/0000-0002-2505-9987
Alan Brown (ID) https://orcid.org/0000-0002-0021-0476

### Decision letter and Author response
Decision letter https://doi.org/10.7554/eLife.53322.sa1
Author response https://doi.org/10.7554/eLife.53322.sa2

# Additional files

## Supplementary files

• Transparent reporting form

## Data availability

The EM density map for the BBSome has been deposited under accession code EMD-21144 and the EM density map for the BBSome:ARL6:GTP complex has been deposited under accession code EMD-21145. Masks and maps from multibody refinement are included as additional maps in these depositions. The corresponding atomic models have been deposited under accession codes 6VBU and 6VBV.

The following datasets were generated:

| Author(s) | Year | Dataset title | Dataset URL | Database and Identifier |
|---|---|---|---|---|
| Singh SK, Gui M, Koh F, Yip MCJ, Brown A | 2020 | Structure of the bovine BBSome (map) | https://www.ebi.ac.uk/pdbe/emdb/EMD-21144 | Electron Microscopy Data Bank, EMD-21144 |
| Singh SK, Gui M, Koh F, Yip MCJ, Brown A | 2020 | Structure of the bovine BBSome (model) | https://www.rcsb.org/structure/6VBU | RCSB Protein Data Bank, 6VBU |
| Singh SK, Gui M, Koh F, Yip MCJ, Brown A | 2020 | Structure of the bovine BBSome: ARL6:GTP complex (map) | https://www.ebi.ac.uk/pdbe/emdb/EMD-21145 | Electron Microscopy Data Bank, EMD-21145 |
| Singh SK, Gui M, Koh F, Yip MCJ, Brown A | 2020 | Structure of the bovine BBSome: ARL6:GTP complex (model) | https://www.rcsb.org/structure/6VBV | RCSB Protein Data Bank, 6VBV |

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
