## [Decision Letter]

**Acceptance summary:**

The manuscript by Brown et al., reports the high-resolution cryo-EM structures of the bovine BBSome with and without ARL6:GTP bound. The structures are significantly higher resolution (3.1-3.5A) than the BBSome cryo-EM structure reported by Chou et al., earlier this year and allow for complete modeling of the BBSome complex including side-chains. Most importantly, the structures elucidate how conformational changes allow the BBSome complex to bind ARL6:GTP as required for membrane association. Additionally, the authors discover density for 2 calcium binding sites in the BBS2 subunit with potential regulatory function. Also, given the higher resolution, the manuscript presents a comprehensive mapping of BBS disease mutations onto the structure.

**Decision letter after peer review:**

Thank you for submitting your article "Structure and activation mechanism of the BBSome membrane-protein trafficking complex" for consideration by *eLife*. Your article has been reviewed by three peer reviewers, and the evaluation has been overseen by a Reviewing Editor and John Kuriyan as the Senior Editor. The following individuals involved in review of your submission have agreed to reveal their identity: Esben Lorentzen (Reviewer #2); Masahide Kikkawa (Reviewer #3).

The reviewers have discussed the reviews with one another and the Reviewing Editor has drafted this decision to help you prepare a revised submission.

Essential revisions:

1) A biochemistry figure or figure supplement is lacking (e.g. showing the quality of the BBSome preparation). This is especially missed in Results paragraph one, where biochemical results are discussed.

2) In the submitted PDB files, the coordinates for the Ca^2+^ ion near BBS2 propeller blade 6 appear to be severely misplaced (cf. Figure 2E).

3) The conformational change coupled with ARL6 binding is arguably the key result of the paper, but I think could be more clearly illustrated in Figure 6. Perhaps a panel plotting RMSD on the structure between states or showing inter-atomic displacement vectors could help depict the motion. The authors may also have their own ideas.

---

## [Author Response]

Essential revisions:1) A biochemistry figure or figure supplement is lacking (e.g. showing the quality of the BBSome preparation). This is especially missed in Results paragraph one, where biochemical results are discussed.

We have added a panel (Figure 1—figure supplement 1a) showing the purity of the BBSome after purification.

2) In the submitted PDB files, the coordinates for the Ca^2+^ ion near BBS2 propeller blade 6 appear to be severely misplaced (cf. Figure 2e).

We apologize for this misplacement, which must have happened as we were preparing the files for submission. We have corrected the positioning in the final models deposited to the PDB.

3) The conformational change coupled with ARL6 binding is arguably the key result of the paper, but I think could be more clearly illustrated in Figure 6. Perhaps a panel plotting RMSD on the structure between states or showing inter-atomic displacement vectors could help depict the motion. The authors may also have their own ideas.

We have added a new figure (Figure 6—figure supplement 1) showing the position of BBS1 in the unbound and ARL6-bound states, as well as two views of the conformational change displayed as inter-atomic displacement vectors.